# Mother and Daughter Carrying of the Same Pathogenic Variant in *FGFR2* with Discordant Phenotype

**DOI:** 10.3390/genes13071161

**Published:** 2022-06-27

**Authors:** Filomena Lo Vecchio, Elisabetta Tabolacci, Veronica Nobile, Maria Grazia Pomponi, Roberta Pietrobono, Giovanni Neri, Simona Amenta, Ettore Candida, Cristina Grippaudo, Ettore Lo Cascio, Alessia Vita, Federica Tiberio, Alessandro Arcovito, Wanda Lattanzi, Maurizio Genuardi, Pietro Chiurazzi

**Affiliations:** 1UOC Genetica Medica, Fondazione Policlinico Universitario “A. Gemelli” IRCCS, 00168 Roma, Italy; filo.flv@gmail.com (F.L.V.); mariagrazia.pomponi@policlinogemelli.it (M.G.P.); simona_am@hotmail.it (S.A.); maurizio.genuardi@unicatt.it (M.G.); 2Dipartimento Universitario Scienze Della Vita e Sanità Pubblica, Sezione di Medicina Genomica, Università Cattolica del Sacro Cuore, 00168 Roma, Italy; elisabetta.tabolacci@unicatt.it (E.T.); veronicanobile88@gmail.com (V.N.); roberta.pietrobono@unicatt.it (R.P.); giovanni.neri43@gmail.com (G.N.); 3Fondazione Policlinico Universitario “A. Gemelli” IRCCS, 00168 Roma, Italy; wanda.lattanzi@unicatt.it; 4Private Practice, 00100 Rome, Italy; ettore.candida@gmail.com; 5Dipartimento Testa Collo, Fondazione Policlinico Universitario “A. Gemelli” IRCCS, Università Cattolica del Sacro Cuore, 00168 Roma, Italy; cristina.grippaudo@unicatt.it; 6Dipartimento di Scienze Biotecnologiche di Base, Cliniche Intensivologiche e Perioperatorie, Università Cattolica del Sacro Cuore, 00168 Roma, Italy; alessandro.arcovito@unicatt.it (A.A.); ettore.locascio@unicatt.it (E.L.C.); 7Dipartimento Universitario Scienze della Vita e Sanità Pubblica, Sezione di Biologia Applicata, Università Cattolica del Sacro Cuore, Largo F. Vito 1, 00168 Roma, Italy; alessia.vita@unicatt.it (A.V.); federica.tiberio@unicatt.it (F.T.)

**Keywords:** *FGFR2*, craniosynostosis, synonymous variant, clinical phenotype, genetic medicine, neurosurgery

## Abstract

Craniosynostosis are a heterogeneous group of genetic conditions characterized by the premature fusion of the skull bones. The most common forms of craniosynostosis are Crouzon, Apert and Pfeiffer syndromes. They differ from each other in various additional clinical manifestations, e.g., syndactyly is typical of Apert and rare in Pfeiffer syndrome. Their inheritance is autosomal dominant with incomplete penetrance and one of the main genes responsible for these syndromes is *FGFR2*, mapped on chromosome 10, encoding fibroblast growth factor receptor 2. We report an *FGFR2* gene variant in a mother and daughter who present with different clinical features of Crouzon syndrome. The daughter is more severely affected than her mother, as also verified by a careful study of the face and oral cavity. The c.1032G>A transition in exon 8, already reported as a synonymous p.Ala344 = variant in Crouzon patients, also activates a new donor splice site leading to the loss of 51 nucleotides and the in-frame removal of 17 amino acids. We observed lower *FGFR2* transcriptional and translational levels in the daughter compared to the mother and healthy controls. A preliminary functional assay and a molecular modeling added further details to explain the discordant phenotype of the two patients.

## 1. Introduction

Craniosynostosis is the consequence of the premature closure of one or more cranial sutures. It is associated with distortion of the skull shape due to the lack of growth perpendicular to the fused suture and compensatory overgrowth at non-fused sites. Its overall prevalence has been estimated at 1:2100–2500 live births, affecting males slightly more often than females [1,2]. Environmental, such as intrauterine fetal head constraint, and genetic factors, including single gene variants, chromosome anomalies and polygenic background, may predispose to craniosynostosis. The condition is more often sporadic, although it can recur in some families due to Mendelian inheritance. Additionally, it can be classified into non-syndromic or syndromic forms, the latter ones being combined with other anomalies of bone differentiation, notably of the hands and feet. Importantly, regardless of the clinical presentation, involvement of a major genetic contribution should be assessed to define recurrence risk in the families [3]. The first description of a genetic origin of syndromic craniosynostosis dates to 1993 by Jabs et al. [4], who identified a variant in the *MSX2* gene in a patient with Boston type craniosynostosis. To date, around 80 genes have been identified as causative of craniosynostosis. Among these, *FGFR2*, *FGFR3*, *TWIST1* and *EFNB1* are the most frequently involved [5,6]. Most genetic forms of craniosynostosis are transmitted as autosomal dominant traits with reduced penetrance. *FGFR2* and *FGFR3* are subject to the paternal age effect: variants in both genes present the gain-of-function effect, positively correlate with elevated paternal age and are positively selected during spermatogenesis [7]. The *FGFR2* gene maps to chromosome 10q26 and is a member of the FGFR family with several isoforms due to alternative splicing [8]. It encodes a tyrosine kinase receptor, which has an extracellular ligand-binding portion composed of three immunoglobulin-like domains (IgI, IgII and IgIII), a transmembrane region, and an intracellular tyrosine kinase domain [9]. Pathogenic variants, mostly recurrent missense substitutions causing gain-of-function, are mainly localized in exon 7 (IgIIIa) and exon 8 (IgIIIc) of the gene [3,10]. The cellular consequences of such variants include enhancement of proliferation, differentiation and apoptosis of osteoblasts bordering the cranial suture mesenchyme; premature differentiation is presumably the most important factor leading to craniosynostosis [11,12]. Monoallelic *FGFR2* variants are associated with three syndromic forms of craniosynostosis, namely Apert (OMIM #101200), Crouzon (OMIM #123500) and Pfeiffer (OMIM #101600) syndromes. All of these syndromes exhibit a distinctive facial appearance, with orbital proptosis, hypertelorism and midface hypoplasia [13,14]. Syndactyly is generally absent in Crouzon syndrome, while it is a distinguishable finding in Apert syndrome [13,14]. The phenotype of Pfeiffer syndrome is characterized by broad thumbs and toes [14]. Affected patients usually do not have intellectual disability. Less frequently, *FGFR2* pathogenic variants may cause other craniosynostosis forms. Jackson–Weiss syndrome (OMIM #123150) presents with foot anomalies, such as broad great toes with medial deviation, broad short metatarsals broad proximal phalanges, and partial syndactyly of the second and third toes [9,15,16]. Beare–Stevenson cutis gyrata syndrome (OMIM #123790) is characterized by a classical crouzonoid face, choanal stenosis or atresia, cutis gyrata and developmental delay [17,18]. Additional findings include umbilical stump, acanthosis nigricans, hirsutism, palatal and genitourinary anomalies (anteriorly placed anus, hypospadias); a high rate of sudden unexplained death was reported [19,20]. Bent-bone dysplasia (OMIM #614592) is a perinatal lethal skeletal dysplasia that exhibits low-set ears, micrognathia and premature eruption of fetal teeth. Bent long bones, osteopenia, irregular periosteal surfaces, low skull ossification and hypoplastic clavicles and pubis are frequently reported by radiographic examinations [21]. According to their monogenic inheritance, some authors propose to group these syndromes in a single spectrum of conditions termed “*FGFR2*-associated craniosynostosis syndromes”, including also clinically intermediate forms [14].

Here we describe a mother and her daughter with a clinical phenotype compatible with Crouzon syndrome, showing different severity, as documented by their clinical history and by a thorough imaging study of the oral cavity and facial measurements. Both patients were heterozygous for the variant c.1032G>A in exon 8 (also called exon 10) of the *FGFR2* gene. The variant, which does not cause coding changes, has already been reported several times in the literature [3,9,13,14,15,22,23,24]. Our results demonstrate an alternative splicing due to this variant, through which a new donor splice site (-GT) is activated with a consequent deletion of 51 nucleotides at the 3′ of exon 8. To address the difference in the clinical phenotype of our patients, relative quantification of *FGFR2* mRNA and Western blot analysis of the FGFR2 protein were performed. Lower levels of *FGFR2* transcription and translation were found in the daughter compared to her mother and healthy controls. To better clarify the effect of this variant in both patients, an osteoinduction assay and a molecular modeling with the mutated and wild-type protein were performed. These results are discussed in light of previous findings reported in the literature.

## 2. Materials and Methods

### 2.1. Patients and Samples

Written informed consent was provided by both mother and daughter to perform molecular analysis as well as for sampling and sharing of clinical information, including pictures. The study protocol was approved by the Ethics Committee of the Catholic University of Rome (prot. N. 9917/15 and prot.cm 10/15). Skin biopsies were performed on both patients to undertake expression studies.

DNA was extracted from peripheral blood leukocytes using the standard salting-out protocol. RNA was extracted from skin fibroblasts using Trizol protocols (Thermo scientific, Waltham, MA, USA). RNA was analyzed to gauge expression levels of *FGFR2* in both patients and four neurotypical controls’ fibroblasts (two females aged 45 and two females aged 20) and of *RUNX2* in patients and two control age-matched fibroblasts.

### 2.2. Cell Cultures and Osteogenic Differentiation Assay

Skin fibroblasts were grown at 37 °C with 5% CO_2_ in DMEM High Glucose (Biowest, Nuaillé, France) supplemented with 10% fetal bovine serum (Biowest, Nuaillé, France), 2.5% L-glutamine (Biowest, Nuaillé, France) and 1% penicillin-streptomycin (Biowest, Nuaillé, France) (complete DMEM medium) and early passages (up to 5th–6th) were employed for the following studies.

For the osteoinduction assay, cells were first grown in complete DMEM medium until reaching confluence. Then, the osteogenic medium composed of DMEM with low glucose (1 g/L) (Aurogene, Rome, Italy) supplemented with 1% L-glutamine, 1% antibiotics (penicillin 100 IU/mL, streptomycin 100 mg/mL) (Euroclone, Milan, Italy), 10% fetal bovine serum (GIBCO by ThermoFisher Scientific, Waltham, MA, USA), dexamethasone (0.1 μM), ascorbic acid (10 μM) and β-glycerophosphate (50 μM) (Sigma Aldrich, Saint Louis, MO, USA) was added. Every 3 days, medium was changed, and cells were detached by trypsin at different time points (24 h, 48 h and 6 days). Cells cultured in complete DMEM medium were used as negative controls for the osteoinduction assay.

### 2.3. Molecular Diagnosis on Genomic DNA

Massive parallel sequencing was performed on the Ion Torrent PGM platform (Thermo scientific, Waltham, MA, USA). A Custom Panel Ion Ampliseq On-Demand IAD208907 was employed that includes 620 amplicons, spanning the exon–intron junctions near the splice sites. Genes included in the panel are listed in Table 1. Data were analyzed through Ion Reporter program version 5.12 (Thermo scientific, Waltham, MA, USA).

The *FGFR2* variant c.1032G>A, located in exon 8, was detected in a heterozygous state and confirmed by Sanger sequencing (access number NM_000141.4 corresponding to t1 transcript); apparently it should not cause an amino acid change p.(Ala344=). The genomic DNA of both patients was amplified by PCR using specific primers (IDT, Leuven, Belgium), as reported in Table 2. The PCR products were purified with exonuclease and alkaline phosphatase (Biotechrabbit GmbH, Berlin, Germany) and sequenced in both directions using BigDye terminator v3.1 (Thermo scientific, Waltham, MA, USA) on a 3130*xl* Genetic Analyzer (Thermo scientific, Waltham, MA, USA).

### 2.4. RNA Analysis

A total of 700 ng of total RNA extracted from fibroblasts was reverse-transcribed into cDNA with a SensiFASTA cDNA Synthesis Kit (Meridian bioscience, Memphis, TN, USA). The cDNA of the patients was PCR amplified with *FGFR2* coding primers, as displayed in Table 2.

Ten different amplifications for both patients were pooled before gel extraction, while a single PCR was sufficient for the control sample. After checking the PCR products on 2% agarose gel, the amplicons were extracted using a StrataPrep DNA Gel Extraction Kit (code 400766, Agilent Technologies, Santa Clara, CA, USA) and cloned using the StrataClone PCR Cloning Kit (code 240205, Agilent Technologies, Santa Clara, CA, USA). White colonies were selected and sequenced using specific plasmid primers T3 and T7.

For relative quantification of the *FGFR2* transcript using the ABI7900HT machine (Thermo scientific, Waltham, MA, USA), the following pre-developed TaqMan^®^ assays were employed: *FGFR2* (Hs.PT.58.1565679, location exons 6-7; IDT, Leuven, Belgium) and *GAPDH* (glyceraldehyde-3-phosphate-dehydrogenase) (Hs.PT.39a.22214836; IDT, Leuven, Belgium), the latter of which was used for normalization. The cycle parameters were 2 min at 50 °C and 10 min at 95 °C, followed by 40 cycles of 15 s at 95 °C (denaturation) and 1 min at 60 °C (annealing/extension). The relative quantification of the target transcript (*FGFR2*) vs. normalizer (*GAPDH*) was calculated as follows: 2^−(Ct(*FGFR2*))−Ct(*GAPDH*))controls vs. patients^ = 2^−ΔΔCt^, where ΔCt is the difference (Ct(*FGFR2*)−Ct(*GAPDH*)) and Ct is the cycle at which the detected fluorescence overcomes the threshold. Each sample was evaluated in triplicate and three independent technical replicates were performed. The expression of the *RUNX2* (Runt-related transcription factor 2) gene was analyzed at each time point (24 h, 48 h and 6 days), using semi-quantitative real-time PCR, as already described [25]. Briefly, qPCR was performed using the GoTaq(R) qPCR Master Mix (Promega) and expression levels were normalized using *ACTB* (β-actin) as a reference housekeeping gene. The sequences of oligonucleotide primers were the following: forward 5′-TCGTGCGTGACATTAAGGAG and reverse 5′-CCATCTCTTGCTCGAAGTCC for *ACTB*; forward 5′-GAACCCAGAAGGCACAGACA and reverse 5′-GGATGAGGAATGCGCCCTAA for *RUNX2*. The relative quantification of target transcript (*RUNX2*) vs. normalizer (*ACTB*) was calculated as follows: 2^−(Ct(*RUNX2*))−Ct(*ACTB*))controls vs. patients^ = 2^−ΔΔCt^, where ΔCt is the difference (Ct(*RUNX2*)−Ct(*ACTB*)) and Ct is the cycle at which the detected fluorescence overcomes the threshold. Each sample was evaluated in triplicate and two independent age-matched controls were included. All variables were analyzed by means of descriptive statistics (mean, median, standard deviation, and standard error of mean) using the GraphPad Prism 7 software (Dotmatics, La Jolla, CA, USA).

### 2.5. Western Blot

Proteins were extracted from skin fibroblasts of an unaffected control and both probands using RIPA buffer with protease inhibitors and two independent WB were performed. Primary antibodies were used at the following concentrations: 1:1000 against-FGFR2 rabbit monoclonal antibody (#11835, Cell Signaling Technology, Danvers, MA, USA) and 1:25,000 against-GAPDH rabbit antibody (Abcam, Cambridge, UK).

### 2.6. Imaging Study

3D stereophotogrammetry and 3D scanning of the oral cavity were used in order to obtain an accurate evaluation without exposing patients to radiation [26]. The .stl files of the face were acquired with stereophotogrammetry (3dMDtrio System; 3dMD, Atlanta, GE, USA), considered the gold standard for stereophotogrammetry [27], and were then processed using the Geomagic 2014 (64 bit) software. To delineate the most peculiar facial signs of Crouzon syndrome, the following linear and surface measurements were carried out: surface measurement of eyeball protrusion (EBP) and linear measurement of inner intercanthal distance (ICD). Patients with craniosynostosis have a narrow and high palatine vault. There are no specific reference parameters in the literature, so we compared the measurements taken on the two patients with values reported for the normal population and for oral breathing patients, who have a similar palate shape [28]. Linear, surface and volumetric measurements of the palatine vault (PV) were calculated by using the intraoral scans of the dental arches and palate (Trios, Model T12P, 3Shape, Copenhagen, Denmark). After obtaining the linear measurement of the palates, the surface of the PV was selected and cut out following the dento-gingival junction of the erupted teeth; the surface areas of the two patients were thus obtained. In addition to the area, an axial and a coronal plane were designed. The axial plane was created passing through the landmarks corresponding to the dento-gingival junction of the erupted teeth, while the coronal plane was designed perpendicular to the first and passing through the most distal part of the palatine vault, corresponding to the distal portion of the second molars. After building the floors, the volume of the PV up to the floor was calculated.

### 2.7. Model Building and Steered Molecular Dynamics (SMD)

The model structure for the FGFR2-FGF1 complex was retrieved from the Protein Data Bank (PDB code: 1DJS) [29] and was designed with Maestro Protein Preparation Wizard [30], using default parameters. The mutated complex FGFR2 p.Ala344 = −FGF1, was prepared by homology modeling using Prime [31], deleting in silico the 17 amino acids after Ala344 within the FGFR2 structure and adding the aminoacidic sequence (APGRE) at the C-terminus of the extracellular domain. Thereafter, the wild type and mutated FGFR2-FGF7 complexes were obtained by superimposing the FGF7 structure (PDB code: 1QQK [32]) to FGF1 in the previously described complexes. The resulting systems, FGFR2 WT–FGF7 and FGFR2 p.Ala344 = −FGF7, were minimized using the Macromodel application of the Schrödinger suite [30].

All of the SMD simulations were performed using GROMACS 2021.4 [33] with the CHARMM36m [34] force-field at the full atomistic level and TIP3 water solvent. The protein-ligand systems were solvated in a cubic water box with a separation margin from the solute of 2.5 nm in each dimension (box size (14.5 × 11.4 × 9) nm) under periodic boundary conditions. The total charge of the system was neutralized by randomly substituting water molecules with Na+ ions and Cl– ions, to obtain neutrality with 0.15 M salt concentration. Following a steepest descent minimization algorithm, the system was equilibrated in canonical ensemble (NVT) conditions for 125 ps, using a Nose–Hoover thermostat with position restraints for the protein–protein complexes. Then, all positional restraints were removed except for the receptor α-carbon atoms (CA), which were used as immobile references. SMD runs were performed under NPT conditions at 303.15 K, using a Nose–Hoover thermostat, with a T-coupling constant of 1 ps, and a Parrinello–Raman barostat at 1 atm. Van der Waals interactions were modelled using a 6–12 Lennard–Jones potential with a 1.2 nm cut-off. Long-range electrostatic interactions were calculated, with a cut-off for the real space term of 1.2 nm. All H-bonds were constrained using the LINCS algorithm. The timestep employed was 2 fs, and the coordinates were saved every ps. The pulling force to unbind FGFs from the receptor was applied along the vector connecting the two protein domains. In particular, the spring constant was set to 1000 KJ mol^−1^ nm^−2^ and the pulling velocity to 0.01 nm/ps.

## 3. Results

### 3.1. Clinical Evaluations

The two probands are a mother and her daughter (Figure 1). The mother was 52 years old, and no information was available on her original family since she was adopted. Her past medical history records that, at the age of 11 years, she wore a brace for 6 months due to scoliosis. An MRI of the vertebral column detected a lumbar hernia. Clinical evaluation showed short stature (155 cm, 10th centile), hypertelorism, and a widened and curved nasal dorsum.

Her daughter is now 20 years old and was born at 38 weeks after an uneventful pregnancy. At birth she weighed 2380 g (3rd–10th centile), length was 45 cm (3rd centile) and head circumference was 33.5 cm (25th–50th centile). No other problems were reported during the neonatal period. At 6 months, following clinical suspicion of craniosynostosis, a skull CT scan was performed. This showed prevalence of the antero-posterior over the biparietal diameter, narrow occipital scale, bilateral lacunar planks in the parietal-occipital area, stenosis of the cranial sutures except for the coronal, dilation of the pericerebral liquor spaces in the frontal convexity and in the medial part of the hemispheres. She therefore underwent a first cranioplasty surgery, followed by two additional interventions at 3 and half and at 13 years of age. She acquired autonomous walking at the age of 14 months and language development was normal. At the age of 5 she had the first genetic consultation at our clinic and on that occasion, following clinical suspicion of Crouzon syndrome, mutation screening of the *FGFR1* and *FGFR2* genes was performed through the dHPLC technique, with normal results. She underwent a second genetic evaluation at the age of 17 years. Currently, her measurements are: height 155 cm (10th–25th centile), weight 52 kg (25th–50th centile) and head circumference 53.5 cm (25th centile). The girl presented with slightly upslanting eyelids, proptosis, straight nose with widened tip, receding forehead, relative hypoplasia of the mandible, reverse bite, high and narrow palate, mid-facial hypoplasia, low-set ears, mild *pectus excavatum*, proximally implanted thumbs, bilateral valgus hallux, and partial bilateral cutaneous syndactyly of the II–III toes. The overall facial and skull phenotype has greatly benefited from the various surgical interventions, including mid-facial advancement at 12 years old. The clinical hypothesis of craniosynostosis remained convincing and thus, molecular investigation of the *FGFR3* and *TWIST1* genes and array-CGH analysis to search for microdeletions/duplications were performed, all with negative results. Eventually, a custom gene panel specific for craniosynostosis was employed for both patients (see Section 2.3).

### 3.2. Imaging Study

Through comparative analysis between the three-dimensional images of the faces and oral cavities of the two patients, linear, surface and volumetric measurements were taken at the age of 20 (daughter) and 52 years (mother) (Figure 2). The surface measurement of the EBP was almost identical: 35,438 mm in the daughter and 35,939 mm in her mother. A greater ICD of the daughter (35,438 mm) than her mother (30,853 mm) was evident. The mean value of the ICD in women is 28.15 ± 2.75 mm [35,36]. Hypertelorism was noted in the daughter upon physical examination, and a wider PV (surface and volumetric measurements) has been observed in the daughter (1716 and 4661.93 mm^3^, respectively) compared with her mother (1360 and 3619.28 mm^3^, respectively). In patients with nasal breathing and normal transverse values of the palate, the palate area and volume corresponded to 923 and to 3756 mm^3^, respectively [28]. These values, compared with those of our patients, highlight how the daughter’s palate, although narrow, results in larger volumes and surface areas, unlike the mother who has a normal volume of the palate and an area of surface close to the norm.

### 3.3. Molecular Investigations

Using a specific custom panel, the single nucleotide variant c.1032G>A (rs121918491) in the *FGFR2* gene was eventually identified in a heterozygous state in both probands, and subsequently confirmed by Sanger sequencing (Figure 3A,B). This genomic change corresponds to the synonymous variant p.(Ala344=), but it was found in just 1 out of 251,454 alleles of the gnomAD database v.2.1.1 (https://gnomad.broadinstitute.org, 1 June 2022) with a minor allele frequency of 3.97 × 10^−6^. It is reported by ClinVar as “pathogenic” (https://www.ncbi.nlm.nih.gov/clinvar/variation/13268, 1 June 2022) [37]. In order to ascertain whether the variant could have an effect on RNA processing, we performed splicing analysis. While cDNA amplification of a control sample yielded a single band of 300 bp, both patients showed an additional band of approximately 250 bp. The lower band was less prominent on agarose gel in the daughter sample compared to the mother, while the upper normal band appeared more evident in the daughter than her mother, although it was weaker than the control in both probands (Figure 3C). Sequencing after cloning of the two bands demonstrated that the upper band corresponded to the wild-type full length allele, while the smaller band lacked 51 nucleotides (Figure 3D). The smaller fragment derived from the activation of a new donor GT splice site as a consequence of the G to A change, leading to skipping of 51 nucleotides at the 3′ end of exon 8 and resuming in frame with the following exon. Though not quantitative, amplifications performed with specific oligo pairs for both alleles (deleted and not; amplicon of 98 bp) and for the non-deleted one (amplicon of 124 bp) showed lower transcript intensity in the two patients for both amplifications, particularly in the daughter compared with two controls (Figure 4A). We quantified the *FGFR2* transcript in the two patients and healthy controls using relative qPCR (Figure 4B). Using a probe that amplifies all isoforms, significantly lower levels of the *FGFR2* transcript were observed in the daughter compared with both her mother and healthy controls. Western blot analysis revealed low FGFR2 levels in both patients compared with a healthy control cell line (Figure 4C). Though protein data are preliminary, they are similar to those observed at the transcriptional level. To further explain how p.(Ala344=) may cause discordant phenotypes in the mother and her daughter, an osteoinduction assay was performed on skin fibroblasts. Fibroblasts derived from both patients and two age-matched controls were cultured in an osteoinductive medium. The expression of the master bone transcription factor RUNX2, selected as a key downstream effector of FGFR2 activation, has been quantified at different time points. *RUNX2* expression in the daughter’s fibroblasts were strongly upregulated during *in vitro* osteogenic induction compared to her mother’s and controls’ cells, after 24 and 48 h (Figure 4D). Conversely, 6 days after bone induction, *RUNX2* expression levels were similar in patients and controls, as before osteoinductive treatment.

### 3.4. Molecular Modeling

To better clarify the pathogenic effect of this variant, a molecular modeling was undertaken. The mutant FGFR2 p.(Ala344=) shows a reduced affinity toward FGF1 compared to the wild type receptor, as its force peak value (orange curve) is 1363 ± 63 KJ mol^−1^ nm^−1^, lower than the FGFR2 WT–FGF1 force peak value (blue curve) that corresponds to 1507 ± 5 KJ mol^−1^ nm^−1^ (Figure 5A). This effect is a consequence of a repositioning of the shortened mutated receptor that reduces its contact surface with its preferred ligand FGF1. Conversely, the mutated receptor increases its affinity for FGF7, an alternative partner that exhibits lower affinity to the wild type receptor, as indicated by the following force peak values: FGFR2 p.(Ala344=) −FGF7 = 1196 ± 32 KJ mol^−1^ nm^−1^ and FGFR2_WT–FGF7 = 1019 ± 16 KJ mol^−1^ nm^−1^ (Figure 5B). This result is due to the formation of a novel strong interaction between the Arg347 of the mutated receptor with Glu141 of FGF7 (Figure 5C,D). In the same topological position, the wild type receptor has a Serine residue that is unable to form this salt bridge. Accordingly, we speculate that the gain of function effect of the mutation is not a direct consequence of the higher affinity for a specific FGF ligand, but rather of an increased affinity for a wider range of FGF ligands [29]. This would allow the receptor to interact more efficiently with a larger set of protein ligands, leading to overall increased activation of the downstream signaling.

## 4. Discussion

The c.1032G>A *FGFR2* synonymous variant detected in this small family has been previously reported in several Crouzon patients [3,9,13,14,15,22,23,24], although its pathogenetic mechanism has never been thoroughly investigated. Extreme phenotypic variability was observed across Crouzon families and in members of the same family in whom it was identified [14,24,38]. The phenotypic effect of this variant varies greatly ranging from minor anomalies (slight hypertelorism and maxillary hypoplasia) to severe findings (dolichocephaly and brachycephaly) [14]. Similar clinical variability was observed in the mother and daughter described here. The mother presented very mild findings, such as hypertelorism and high arched palate, compared to her daughter, who underwent four corrective surgeries due to craniosynostosis. Unfortunately, no further clinical information could be obtained on the mother’s family, nor segregation analysis could be undertaken, since she was adopted.

Two previous studies showed that the c.1032G>A variant can cause alternative splicing [22,23]. In both papers the authors demonstrated the presence of an alternative splicing due to the G to A transition in position 1032 with consequent activation of a new donor splice site immediately after the variant, although they did not show any nucleotide sequence in their papers. Conversely, we cloned and sequenced the shorter PCR product of our patients and proved that the alternatively spliced transcript loses 51 nucleotides, potentially resulting in the production of a shorter protein lacking 17 aminoacids. A similar effect on splicing was demonstrated for other synonymous *FGFR2* variants, such as p.(Pro361=), that is associated with mild Crouzon phenotype and reduced penetrance of craniosynostosis [39]. Though our variant affects the immunoglobulin-like (Ig) IIIc domain of the FGFR2 protein, where most pathogenic *FGFR2* variants are located, little is known about its effects on transcription and translation. When amplifying with different oligo pairs specific to the coding sequence, the intensity of amplicons in both probands was weaker than in controls. One pair, which included a forward primer located within the deleted coding portion of exon 8, yielded a 124 bp amplicon that was specific to the full-length wild-type product. The other pair amplified a 98 bp fragment that included both the deleted and the non-deleted alleles, because the reverse primer was designed in the 5′ end of exon 8 prior to the G>A transition. Interestingly, for both amplification products, the intensity of the daughter’s bands was very faint, suggesting an overall lower quantity of the *FGFR2* transcript. Though these experiments were semiquantitative, they were confirmed by qPCR (Figure 4B) that showed significantly lower levels of transcription in the daughter compared to her mother and unaffected controls, using a probe specific to all *FGFR2* isoforms. These results were preliminarily confirmed at the translational level, showing lower FGFR2 protein in the more severely affected daughter. These observations seem to suggest a loss-of-function mechanism, while most *FGFR2* variants have a gain-of-function effect. Therefore, we performed an osteoinductive treatment on patients’ fibroblasts and found that, despite the lower FGFR2 levels, *RUNX2* expression levels were upregulated in the more severely affected daughter after stimulation of the FGFR2 pathway. *RUNX2* is a downstream target of activated FGFR2 and it encodes a key transcription factor associated with osteoblast differentiation. This apparently paradoxical effect (lower protein levels with overactive FGFR2 pathway) may be explained supposing that the aberrant protein, lacking 17 amino acids (from 345 to 361) in the immunoglobulin-like IIIc domain, has a dominant gain-of-function effect on protein function and leads to overactivation of the downstream FGFR2 pathway. We hypothesize that the severe phenotype observed in the daughter may be due to variability in splicing. Furthermore, we cannot exclude that other modifier factors, i.e., other genes and proteins involved in the FGFR2 pathway (or influencing it), may modulate the clinical phenotype. Finally, it is worth remembering that our experiments were conducted on skin fibroblasts; these cells are easy to obtain, but they differ from mesenchymal stem cells of the membranous neurocranium.

Further molecular studies, particularly on the FGFR2 pathway, are warranted to better understand the multiple effects of the c.1032G>A variant, which is definitely not an innocent synonymous change [40].

## Figures and Tables

**Figure 1 genes-13-01161-f001:**
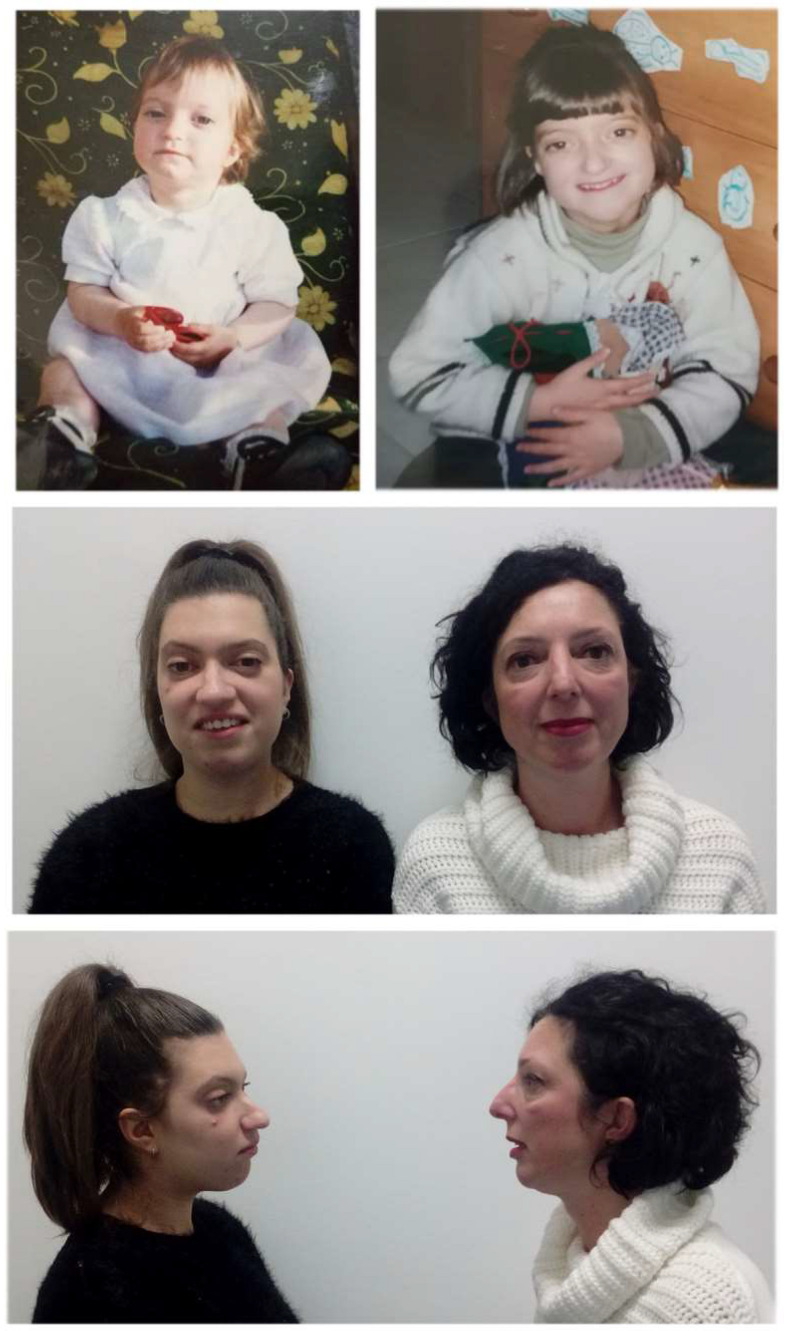
Pictures of the two patients reported in this paper. In the upper panels the daughter was 13 months (left) and 5 years (on the right) old. The crouzonoid face was already evident as a child. In the middle and bottom panels: the daughter after four corrective surgeries at the age of 20 years, and the 52 years old mother in a front and side image, respectively. Note, particularly in the girl, the straight nose with widened tip, receding and square forehead, reverse bite, and low-set ears (in both).

**Figure 2 genes-13-01161-f002:**
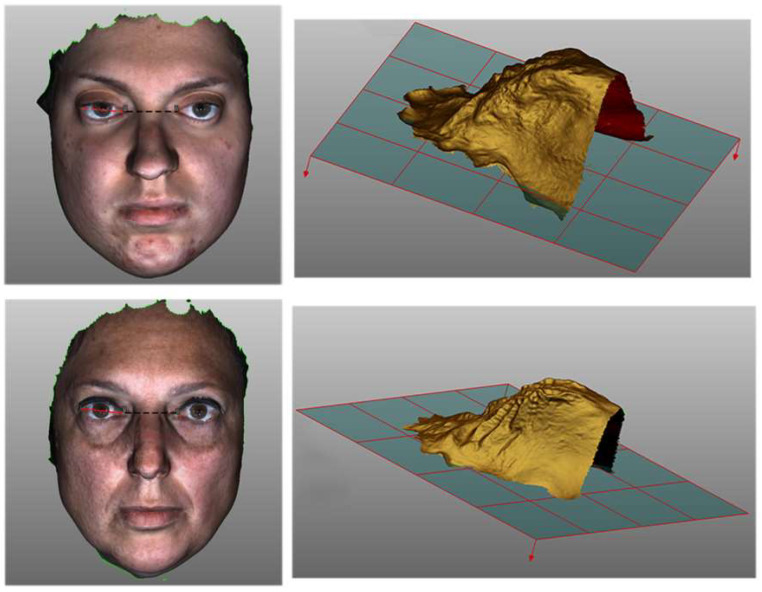
Three-dimensional images of the faces and oral cavities of the daughter at 20 years of age (upper panels) and her mother at 52 years of age (bottom panels). The surface measurement of the EBP (dashed red line) and the linear measurement of the ICD (dashed black line) were obtained by the left images. The right panels represent the 3D images of the volume of oral cavity of the daughter (top) and of the mother (bottom).

**Figure 3 genes-13-01161-f003:**
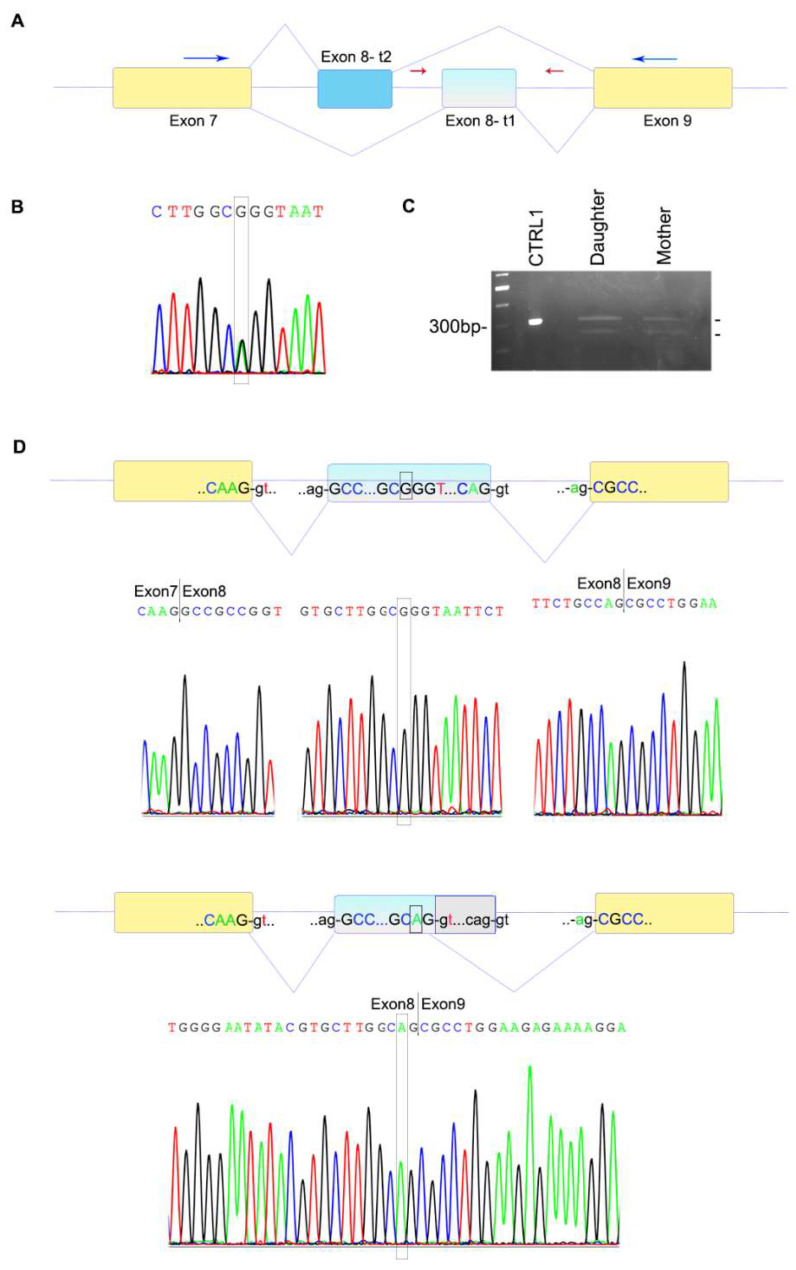
Genomic and coding sequence of the variant. (**A**) Schematic representation of the *FGFR2* gene with exon 8 in the two alternative isoforms (t1 and t2). The yellow boxes represent common exons (number 7 and 9) to the t1 and t2 isoforms of *FGFR2*. The upper lines display the splicing of the t2 isoform, which includes the first exon 8 (in blue). The lower lines depict the splicing of the t1 isoform with the inclusion of the second exon 8 (in light blue). Red arrows represent the position of the genomic primers, while blue arrows those of coding oligos employed in cloning. (**B**) Sanger sequencing of the *FGFR2* exon 8 (t1 isoform) confirms the presence of the heterozygous variant c.1032G>A in both patients (box). This sequence corresponds to the daughter’s Sanger sequence, which is identical to her mother’s. (**C**) Visualization on 2% agarose gel of cDNA amplification products obtained using primers located within the coding sequence (blue arrows in A, coding FL-F and R) on cDNA of a control sample (single PCR) and of both patients (10 PCRs were loaded together into a wider well to better visualize their cDNA products). Note the two bands in both patients. The upper band of 300 bp corresponds to the full-length allele, while the lower band of approximately 250 bp is found only in the two patients. The lower band is weaker in intensity in the daughter compared to her mother. (**D**) Sanger sequencing after cloning of the two bands found in the patients. The coding sequence of the normal allele displayed the presence of G (box) and a normal splicing that entirely included exon 8, continuing to exon 9 (yellow box) (upper panel). The allele carrying the G>A variant was alternatively spliced, lacks the 3′ end of exon 8 (grey box) and is directly spliced to the next exon (bottom panel), skipping 51 nucleotides. Coding and non-coding nucleotides are indicated in upper- and lowercase, respectively.

**Figure 4 genes-13-01161-f004:**
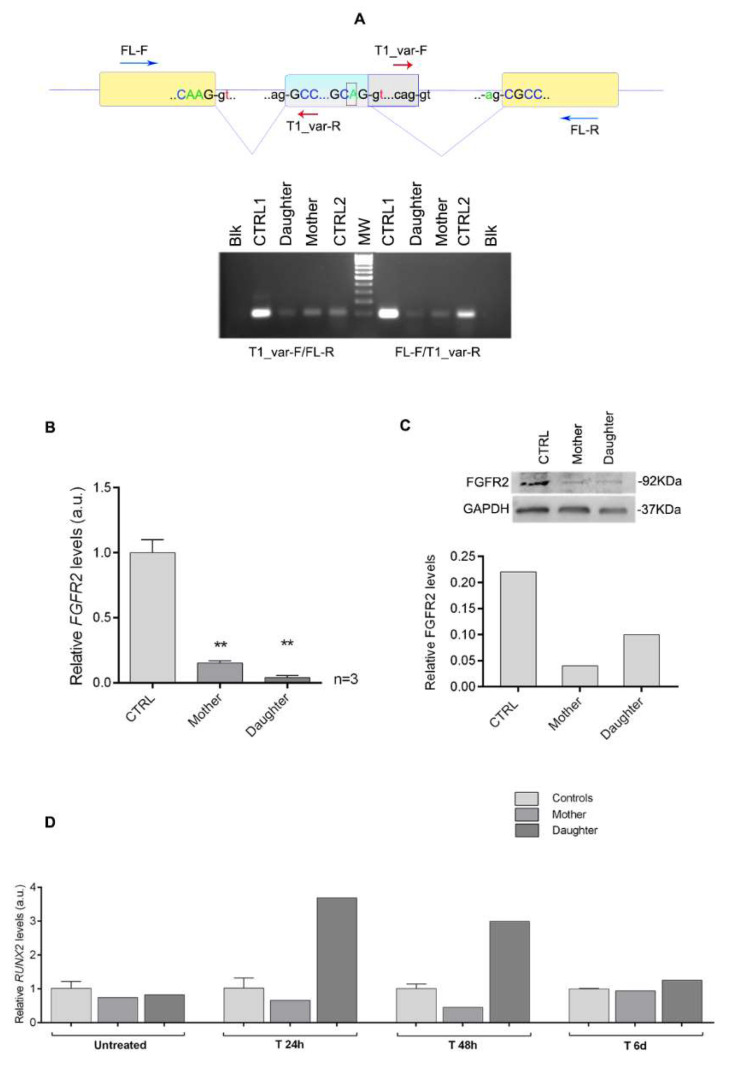
Quantification of the *FGFR2* transcript and protein and of the *RUNX2* transcript after osteoinduction. (**A**) Visualization on 2% agarose gel of PCR amplification of cDNA performed in two control samples and in the patients. On the left of the molecular weight marker, an amplification product of 124 bp obtained using the coding T1_var-F (red arrow on the upper panel) combined to a coding FL-R primer (blue arrow on the upper panel) is visible; this includes only the non-deleted isoforms because the forward primer (red arrow) is located within the deleted portion of exon 8. On the right of the marker, the 98 bp-amplicon obtained with coding FL-F (blue arrow) combined with a coding T1_var-R (red arrow) primer is shown; this encompasses all isoforms (deleted and not) of the *FGFR2* transcript because both primers are placed in the non-deleted mRNA portions. Note the lower intensity of amplification in the patients (more pronounced in the daughter) compared with two normal control samples. Blk = blank (no cDNA). (**B**) Relative quantification of *FGFR2* transcript by real-time RT-PCR on skin fibroblasts of the patients (mother and daughter) and four normal controls. Levels of *FGFR2* RNA were reduced in both probands (particularly in the daughter) compared to controls. Values reported on the y-axis represent relative transcriptional levels normalized to the endogenous transcript GAPDH (2^−Δct^) in the patients vs. controls (2^−ΔΔct^). The quantification was repeated three independent times (*n* = 3). One-way ANOVA was applied to determine statistical significance. ** *p* < 0.05. (**C**) Representative Western blot with antibodies against FGFR2 (92 kDa) and GAPDH (37 kDa) on the protein lysates extracted from a healthy control and fibroblasts from the two patients (on the left). The graph represents relative protein quantification expressed as the ratio between FGFR2 and GAPDH levels. FGFR2 level appears lower in both females compared to an unaffected control, with lower levels in the mother. (**D**) Relative quantification of the *RUNX2* transcript by real time RT-PCR on skin fibroblasts of the patients (mother and daughter) and two normal controls before and after osteoinduction. Levels of *RUNX2* RNA were high in the daughter compared to controls and her mother 24 and 48 h after induction. Values reported on the y-axis represent relative transcriptional levels normalized to the endogenous transcript *β-actin* of patients vs. controls (2^−ΔΔct^). The quantification was performed in triplicate on one experiment of induction.

**Figure 5 genes-13-01161-f005:**
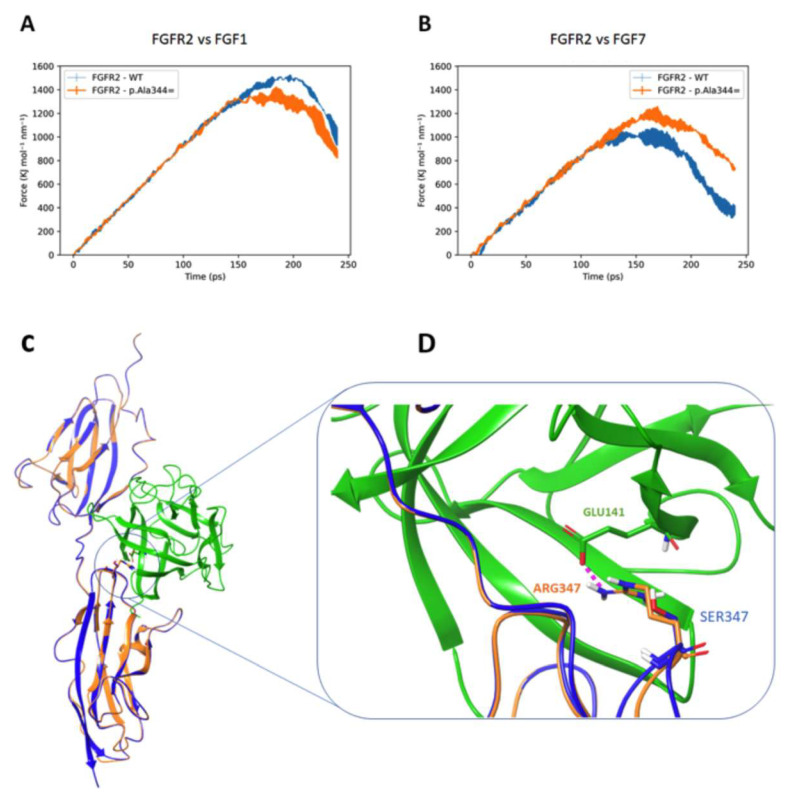
Potential of mean force (PMF) dispersion profiles obtained by 3 SMD replicas for each system. (**A**) Complexes FGFR2 (WT and mutant)—FGF1, blue and orange curves, respectively. (**B**) FGFR2 (WT and mutant)—FGF7, blue and orange curves, respectively. (**C**) Superimposition of FGFR2_WT (blue) and FGFR2_p.(Ala344=) (orange) binding FGF7 (green). (**D**) Magnification of protein–protein interaction: FGFR2 p.Ala344= has an Arginine at topological position 347 (where the WT protein has a Serine) enabling the formation of a salt-bridge interaction with FGF7 Glu141.

**Table 1 genes-13-01161-t001:** List of genes analyzed by massive parallel sequencing with their access number and percentage of coverage.

Gene	Access Number	% Coverage
*SKI*	NG_013084.1/NM_003036.3	55.00%
*WDR35*	NG_021212.1/NM_001006657.2	97.50%
*IFT122*	NG_023392.1/NM_052989.2	100.00%
*ZIC1*	NG_015886.1/NM_003412.4	100.00%
*FGFR3*	LRG_1021-t1 and t2/NM_000142.5 and NM_022965.4	95.00%
*WDR19*	NG_031813.1/NM_025132.4	100.00%
*SPRY4*	NG_034148.1/NM_030964.5	100.00%
*MSX2*	NG_008124.1/NM_002449.5	100.00%
*RAB23*	NG_012170.1/NM_016277.5	100.00%
*RUNX2*	NG_008020.2/NM_001024630.4	79.50%
*GLI3*	NG_008434.1/NM_000168.6	100.00%
*POR*	NG_008930.1/NM_000941.3	95.00%
*FGFR1*	LRG_993-t1/NM_023110.3	100.00%
*RECQL4*	LRG_277-t1/NM_004260.4	95.50%
*FREM1*	NG_017005.2/NM_144966.7	100.00%
*IL11RA*	NG_028966.1/NM_001142784.3	100.00%
*FGFR2*	LRG_994-t1 and t2/NM_000141.5 and NM_022970.3	100.00%
*ALX4*	LRG_1256-t1/NM_021926.4	100.00%
*TCF12*	NG_033851.2/NM_207036.2	100.00%
*SOX9*	NG_012490.1/NM_000346.4	83.50%
*COMP*	NG_007070.1/NM_000095.3	100.00%
*ERF*	NG_042802.1/NM_006494.4	98.50%
*MEGF8*	NG_033030.1/NM_001271938.2	99.00%
*BMP7*	NG_032771.1/NM_001719.3	100.00%
*EFNB1*	NG_008887.1/NM_004429.5	100.00%

**Table 2 genes-13-01161-t002:** Sequence of primers used to amplify the *FGFR2* gene in genomic and complementary DNA.

Primer	Sequence
Genomic F	5′-CCTCCACAATCATTCCTGTGTC
Genomic R	5′-ATAGCAGTCAACCAAGAAAAGGG
Coding FL-F	5′-ACGTGGAAAAGAACGGCAG
Coding FL-R	5′-CACCATACAGGCGATTAAGAAG
Coding T1_var-F	5′-CTTTCACTCTGCATGGTTGA
Coding T1_var-R	5′-CTCAATCTCTTTGTCCGTGG

## Data Availability

The authors confirm that the data supporting the findings of this study are available within the article.

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
