# Peer review of "Mother and Daughter Carrying of the Same Pathogenic Variant in FGFR2 with Discordant Phenotype"

_genes, 2022, doi:10.3390/genes13071161_

Round 1
Reviewer 1 Report
Since the daugther had several surgical interventions it is inappropriate to compare and draw conclusions from the imaging study.
Table 1: NM_# should be given for all genes investigated.
Figure 3 is very difficult to follow. Clearly indicate t1 and t2. In 3C gel pockets appear to have different width. Thus a comparison and conclusion of lower intensities in daughter an mother in relation to the CTRL1 is not possible. Complete sequences incl. WT sequences should be indicated to visualize the splicing effect (figure 3D).
Introduction: "Apert syndrome type 2" - can you provied an OMIM entry or other reference?
your hypothesis "pathogenetic mechanism associated with lower expression of FGFR2" is in conflict with the established pathomechanism of FGFR2 related craniosynostosis i.e. constitutively active receptor/gain-of-function variants. Is receptor dimerization induced by the variant to be expected, molecular modeling? Cite and discuss previous reports (Fenwick et al., Robertson et al. 1998).
Since mRNA was available of mother and daugther downstream FGFR2 targets like RUNX2 should be investigated to demonstrate a misregulation.
Author Response
Reviewer 1
- We cannot change the fact that the daughter required surgical interventions during childhood while her mother facial phenotype was milder. Despite all the surgery, the daughter’s phenotype is still more severe as judged by imaging study (Figures 1 and 2).
- The NM_# access number of the lacking genes was added in Table 1.
- Figure 3 was indeed difficult to follow in its first version. We therefore modified it highlighting the deleted portion of exon 8 in t1 isoform, indicating the partial sequence across splicing junctions for the normal and deleted alleles. We explained in the text and in the figure legend that gel pockets in Figure 3C were wider in order to allow loading 10 PCR reactions of both patients given their lower FGFR2-mRNA levels.
- OMIM entry of Apert syndrome type 2 is the same of Apert and it was repeated accordingly.
- We agree with the Reviewer that lower levels of FGFR2 transcript and protein are in conflict with the established gain-of-function mechanism of Crouzon syndrome. Therefore, we performed a molecular modelling of the in-frame deleted protein and compared its binding efficiency to major FGF ligands (i.e. FGF1 and FG7). We also accepted the Reviewer’s suggestion of investigating RUNX2 as a downstream FGFR2 target with and without osteoinductive treatment. We were delighted to find that RUNX2 is strongly upregulate din the more severe affected daughter compared to her mother and normal controls. Finally, we included Fenwick et al., 2014 in the discussion, but Robertson et al., 1998 was excluded since our variant results in an in-frame deletion downstream of Cysteine 342, involved in the last disulphide bound of the third Ig-like domain.
Reviewer 2 Report
In this manuscript, the authors report an FGFR2 variant in a mother and daughter who present with different clinical features. This synonymous variant has been previously reported in patients with Crouzon syndrome, potentially causing alternative splicing. In this study, the clinical evaluations and image study were performed, and the phenotypes of the two patients with the same variant were compared. The alternatively spliced transcript was identified, consistent with the previous studies. The authors measured the expression of FGFR2 in cultured skin fibroblasts derived from the patients and controls, and observed lower expression of FGFR2 in the daughter compared to the mother and controls. This study provides new insights into the multiple effects of the c.1032G>A variant. The manuscript is well written and organized. However, the quality of some data needs to be improved to support the conclusion. It might be an important conclusion that the c.1032G>A variant could alter the expression of FGFR2 potentially affecting the phenotypes, but the expression analyses are preliminary.
Specific comments:
1. Line 109, were the fibroblasts passaged and maintained before the expression studies?
2. Figure 2: Lines and dash lines in the left panel should be thicker to be better visible. For the right panel, a control image (a 3D image of a normal oral cavity) would be helpful to demonstrate the high arched palate in both patients. Alternatively, I suggest adding arrows to indicate the phenotype.
3. Figure 3B, is the result from the mother or the daughter? I understand the variant is the same, but both sequencing results should be shown.
4. Figure 3C, the lower band for Daughter is vague and inconclusive. A gel image with higher quality is required.
5. Figure 3 and Figure 4. It would be helpful to indicate the lacked region in the patients in the gene structure scheme.
6. Figure 4A, Control 2 is weaker than Control 1 in the gel. Are they controls of different ages? I would suggest defining the four controls with age and other available information in Materials and Methods and specifying which control was used in the experiment. Label the primers with their names in the upper panel.
7. Figure 4B. The controls are analyzed and shown as one sample. How were the four control samples analyzed and quantified? Mixed, combined, or averaged? More details of the method should be provided.
8. Figure 4C. There are four control fibroblast lines, but only one control was used for Western blot. Which control was used (aged 45 or 20)? It would be better to use all or at least two control samples of different ages to evaluate the variation of FGFR2 expression among individuals. How was the protein level quantified? Replicates are required for statistical analysis.
9. The expression studies were performed with skin biopsy-derived fibroblasts. How this cell type is relevant to the development of the phenotypes can be discussed.
Author Response
Reviewer 2
We agree with the Reviewer that the c.1032G>A variant was never thoroughly studied and since our expression data were preliminary, we included further co-authors in order to perform molecular modelling and functional investigation of the FGFR2 pathway. Thanks to these further experiments we now show the altered binding of the deleted protein with its main FGF ligand (molecular modelling) and demonstrate the upregulation of RUNX2 after osteoinduction of patients’ fibroblasts.
- After skin biopsies fibroblasts were maintained and passaged before expression studies, as now specified in Materials and Methods (line 109).
- Lines and dash lines in the left panel of Figure 2 are now thicker. In literature reference measurement for the palatal height is not reported and thus, the description of the palatal height was deleted. We report the values of surface and volumetric measurements, comparable with those described in ref. 27.
- Though both patients were sequenced, in Figure 3B we showed only daughter’s Sanger sequence. We now specified it in Figure legend, as suggested by the Reviewer.
- We acquired a better image of Figure 3C, but the Reviewer has to consider that mRNA levels of the daughter were extremely low.
- Now the in-frame deletion of aminoacids 345 to 361 has been highlighted in gray in both Figures 3 and 4.
- Information on control samples has been included in Materials and Methods. All controls were age-matched. As suggested, we labelled all primers in the upper panel of Figure 4A.
- The four control samples of Figure 4B were combined.
- We agree with the Reviewer that more WB experiments could have been performed on different protein lysates for statistical analysis. We actually did two WB replicates on the same protein lysates. Densitometric analysis of the protein bands was performed to obtain a semi-quantitative estimate of the FGFR2 levels. We may point out that FGFR2 WB analysis is now less relevant after we obtained functional evidence of FGFR2 pathway upregulation.
- As suggested, we discussed the similarities and differences between fibroblasts and mesenchymal stem cells at the end of the Discussion.
Round 2
Reviewer 1 Report
Overall, my previous comments have been adressed and the manuscript has improved by adding the analysis of downstream targets and the molecular modeling to speculate if FGFR2-ligand interaction could be affected.
A few minor comments on the revised version:
Apert syndrome type 2 (ASC II) is outdated nomenclature. Please remove.
Figure 1, bottom panel. low set ears in both mother and daugther.
Figure 3 & 4A were improved significantly.
Figure 4B, D. Different way of visualizing the real time RT-PCR results is irritating. qPCR RUNX2: since each sample was evaluated in triplicate an error bar should be indicated as provided for the controls.
Section 3.4 - references should be added to support the statements regarding FGFR2 and ligand specificity.
Author Response
Reviewer 1
- We removed “Apert syndrome type 2” from the Introduction.
- We agree with the Reviewer that “low set ears” were present in both patients, thus we accordingly added in Figure 1.
- We changed the graph in Figure 4B adapting with that of Figure 4D. Data were expressed as 2-ΔΔct in y axis in both graphs. We performed technical triplicates of qPCR of each sample and their mean was expressed as a single value. For controls we have two different control cells independently treated with osteoinduction medium and the error bar makes sense (two biological replicates).
- Reference 29 (Stauber et al., 2000) was already included in Section 2.7 of the Materials and Methods to support the statements regarding FGFR2 and its ligands. Now it was also added in Section 3.4 of the Results, as suggested.
Reviewer 2 Report
The manuscript has been improved significantly.
I have two additional comments regarding the new data:
Line 418, “a loss of affinity” is not accurate, as it is just relatively lower. It should be “reduced” or “decreased”.
Figure 4, to demonstrate the osteoinduction assay was successful, the authors should show the expression of RUNX2 was increased in the control during differentiation. At least three independent experiments are expected for quantification.
Author Response
Reviewer 2
- We agree with Reviewer and modified the expression “a loss of affinity” in line 418 with “reduced affinity”.
- We expressed RUNX2 levels as 2-ΔΔct, arbitrarily considering control cells as unit. This method does not allow to observe the increase of RUNX2 during differentiation in control cells, although it can be guessed in the graph 4D after 6 days of osteoinduction treatment. We consider preliminary the experiment of differentiation (as specified also in the Abstract) and are aware that three independent experiments represent the gold standard for any quantification.